# Are Early or Late Maturers Likely to Be Fitter in the General Population?

**DOI:** 10.3390/ijerph18020497

**Published:** 2021-01-09

**Authors:** Alan M. Nevill, Yassine Negra, Tony D. Myers, Michael J. Duncan, Helmi Chaabene, Urs Granacher

**Affiliations:** 1Faculty of Education, Health and Wellbeing, University of Wolverhampton, Walsall WS1 3BD, UK; a.m.nevill@wlv.ac.uk; 2Research Unit (UR17JS01) Sports Performance, Health & Society, Higher Institute of Sport and Physical Education of Ksar Saîd, Universite de la Manouba, Tunis 2010, Tunisia; yassinenegra@hotmail.fr; 3Department of Social Science, Sport and Business, Sport and Health, Newman University, Birmingham B32 3NT, UK; Tony.Myers@staff.newman.ac.uk; 4Centre for Sport, Exercise and Life Sciences, Coventry University, Coventry CV1 5FB, UK; michael.duncan@coventry.ac.uk; 5Division of Training and Movement Sciences, Research Focus Cognitive Sciences, University of Potsdam, Am Neuen Palais 10, 14469 Potsdam, Germany

**Keywords:** youth, fitness tests, allometry, body shape, biological age

## Abstract

The present study aims to identify the optimal body-size/shape and maturity characteristics associated with superior fitness test performances having controlled for body-size, sex, and chronological-age differences. The sample consisted of 597 Tunisian children (396 boys and 201 girls) aged 8 to 15 years. Three sprint speeds recorded at 10, 20 and 30 m; two vertical and two horizontal jump tests; a change-of-direction and a handgrip-strength tests, were assessed during physical-education classes. Allometric modelling was used to identify the benefit of being an early or late maturer. Findings showed that being tall and light is the ideal shape to be successful at most physical fitness tests, but the height-to-weight “shape” ratio seems to be test-dependent. Having controlled for body-size/shape, sex, and chronological age, the model identified maturity-offset as an additional predictor. Boys who go earlier/younger through peak-height-velocity (PHV) outperform those who go at a later/older age. However, most of the girls’ physical-fitness tests peaked at the age at PHV and decline thereafter. Girls whose age at PHV was near the middle of the age range would appear to have an advantage compared to early or late maturers. These findings have important implications for talent scouts and coaches wishing to recruit children into their sports/athletic clubs.

## 1. Introduction

The assessment of physical fitness (PF) in children and adolescents is commonplace in schools and sports clubs as part of physical education and long-term athlete development programming [1]. Such testing is typically used to either understand where an individual child performs in relation to a relative group/against normative values for a population or, as a means to assist in intervention targeting. A key and known consideration when trying to understand the development of PF in children and youth is growth and maturation [2]. Timing and tempo of growth and maturation are characterized by large inter-individual variation and so is physical and athletic development [3,4]. This is due to the fact that chronological age and biological maturity seldomly progress at the same rate. [4] The situation becomes even more complex when taking the dynamic interaction between technical, physical, and environmental factors into account that contribute to selection in professional sports academies [5] and influence success in movement-related activity, relative to each child’s age group [6].

An acknowledgement that PF and performance outcomes in children and adolescents are substantially confounded by biological maturation is not new [7,8]. The literature relating to the effect of maturation on children and youth has largely focused on specifically trained populations of youth with an emphasis on sport performance and selection/deselection into academies within long-term athlete development models [9,10]. Relatively fewer studies have examined the effects of maturation on non-specifically trained schoolchildren. In the context of PF, this is important as understanding how maturation might influence PF can be used to inform intervention design and policy for youth related to health improvement and physical activity promotion. There is debate in regard to early sport specialisation in children, and an acknowledgement that engaging in a wider range of sports activity during childhood and youth is essential in providing the building blocks to lead a physically active life and to prepare children to engage in multiple sports as part of the Athletic Skills Model [2]. Thus, examining the association between maturation and fitness in children who are not engaged in high-performance sport is essential in providing information on what might be considered “normal” development of the PF-maturation relationship during childhood and adolescence.

Surprisingly, few studies have examined the relationship between maturation and PF in non-specifically trained children. There is some data which suggests that strategies undertaken by boys in 30 m sprint running are influenced by maturity status, with boys’ pre-peak height velocity (PHV) being more reliant on a step frequency strategy while those post-PHV relied more on a step-length strategy. Studies with children at a national level for basketball demonstrated that differences in PF are associated with maturation status [11,12] and other work has examined how maturation influences PF in the context of relative age effects in soccer academies [5]. There is also data suggesting that maturation influences functional movement in children and youth in soccer academies [13] and high-performance youth athletes [14]. Moreover, the majority of prior studies have tended to examine differences in outcome variables as a consequence of being in different stages of maturation, ignoring the fact that maturation is not a simple linear process. Such approaches do not necessarily consider the continuum of maturation as a dynamic and non-linear process [15]. Analysing paediatric groups as pre- and post-PHV may not enable as nuanced an approach as is needed to fully understand how maturation might influence fitness performance in children and adolescents.

Thus, the present study aimed to examine the association between biological maturation and PF as a continuous process in a sample of non-specifically trained schoolchildren.

## 2. Materials and Methods

### 2.1. Experimental Approach to the Problem

This cross-sectional study was conducted between January to March 2018. It aimed at determining the influence of anthropometric characteristics and biological maturation on PF variables in a sample of untrained schoolchildren. Anthropometric characteristics were collected for all participants in addition to nine PF tests. These included sprint speeds recorded at 10, 20 and 30 m, two vertical jump tests (squat jump [SJ] and countermovement jump [CMJ]) and two horizontal jump tests (standing long jump [SLJ] and the five-jump test [5JT]), the Illinois change of direction test (ICoDT) and handgrip strength test. Tests were always conducted during physical education classes (2 h per week) by the same test instructor.

### 2.2. Participants

The sample consisted of 597 Tunisian children (396 boys and 201 girls) aged 8–15 years. The participants were grouped by chronological age into 1-year age categories. For instance, the group of 8-year-old children included all those aged from 8.00 to 8.99 years (Table 1). In accordance with Lloyd et al. [16], we defined the following sex specific age ranges for children and adolescents. The term “children” refers to girls and boys, generally up to the age of 11 and 13 years, respectively. The term “adolescence” refers to girls 12–18 years and boys 14–18 years.

Besides chronological age, biological maturity was estimated using the maturity offset (MO) method. For both sexes, MO was calculated by predicting age at peak-height-velocity using the predictive equations established by Moore et al. [17]. For girls: MO = −7.709133 + (0.0042232 × age ∗ height). For boys: MO = −7.999994 + (0.0036124 × age ∗ height).

To be enrolled in this study, participants had to be apparently healthy (no diagnosed disease or injury) and not engaged in any structured training program. In this sense, children who were members of sport clubs were excluded from study participation. All schoolchildren were from families of low-to-medium socioeconomic status and enrolled in public elementary or high schools in an urban area of Tunisia.

Written informed parental consent was obtained prior to the start of the study in addition to permission from school authorities. All participants and their parents/legal representatives were fully informed about the experimental protocol and its potential risks and benefits. The study was conducted in accordance with the Declaration of Helsinki, and the protocol was approved by the Ethics Committee of the higher institute of sports and physical education of Ksar Saïd (UR17JS01).

### 2.3. Procedures

#### 2.3.1. Anthropometric Measurements

All the anthropometric measurements were recorded in accordance with standardized procedures of the international society for the advancement of kinanthropometry (ISAK) [18] (Table 1). Each individual’s height (m) and body-mass (kg) were assessed to the nearest 0.1 cm and 0.1 kg, using a SECA stadiometer and a SECA weighing scale (SECA Instruments Ltd., Hamburg, Germany). Body mass index (BMI) was calculated using body-mass divided by height squared (kg/m^2^). All anthropometric measurements were recorded twice by two raters, and the mean scores were retained for further statistical analysis. Inter-rater reliability was assessed using intraclass correlation coefficients (ICC) and standard error of measurement (SEM). For ICCs values were >0.90 and for SEM < 5%.

#### 2.3.2. Sprint Speed

Thirty-meter sprint performance was assessed at 10, 20 and 30 m intervals using an electronic timing system (Microgate, Bolzano, Italy). Participants started in a standing start 0.3-m before the first infrared photoelectric gate, which was placed 0.75-m above the ground to ensure it captured trunk movement and avoided false signals via limb motion. The intraclass correlation coefficients (ICCs) for test-retest reliability ranged between 0.90–0.96 for 10, 20, and 30 m.

#### 2.3.3. Jump Performance

For SLJ, participants started from a standing position with feet shoulder-width apart behind a starting line and arms loosely hanging down. On the command ready, set, go, participants executed a countermovement with their legs and arms and jumped at maximal effort in the horizontal direction. Participants had to land with both feet at the same time and were not allowed to fall forward or backward. The horizontal distance between the starting line and the heel of the rear foot was recorded via tape measure to the nearest 1-cm. The ICC for test-retest reliability was 0.96.

Regarding 5JT, from an upright standing position with both feet flat on the ground, participants tried to cover as much distance as possible with five forward jumps by alternating left- and right-leg ground contacts. The covered distance was measured to the nearest 1-cm using a tape measure. The ICC for test-retest trials was 0.96.

During SJ, participants started from a stationary semi-squatted position with their hands on the iliac crest jumped upward as high as possible. Squat jump performances were recorded through an Optojump photoelectric cell. The ICC for test-retest trials was 0.96.

In terms of CMJ, participants started from an upright erect standing position and performed a fast downward movement by flexing the knees and hips which were immediately followed by a rapid leg extension resulting in a maximal vertical jump. Throughout the execution of the test, participants maintained their arms akimbo. Jump height was recorded using an Optojump photoelectric system. The ICC for test–retest reliability was 0.95.

#### 2.3.4. Handgrip Strength

The child stays in a standard bipedal position with the arms in complete extension holding the dynamometer (TKK 5101; Takei, Tokyo, Japan) without touching any part of the body with it. The dynamometer is adjusted to sex and hand size for each child [19]. Handgrip strength was measured with the subject in a standing position with the shoulder adducted and neutrally rotated and arms parallel but not in contact with the body. The participants were asked to squeeze the handle for a maximum of 3–5 s.

#### 2.3.5. Illinois Change of Direction Test

The dimensions and route directions for the ICoDT were applied in accordance with established methods [20,21]. The performance times were recorded using an electronic timing system and analysed as average speeds given the total distance was taken as 60 m. The ICC for test–retest trials was 0.94.

### 2.4. Statistical Analyses

To identify the most appropriate body-size, shape and MO characteristics (as well as any categorical differences of sex, age, etc.) associated with a variety of physical performance variables, we adopted the following multiplicative model with allometric body size components (Equation (1)). The model is similar to that used to predict the physical performance variables of Greek [22] and Peruvian children [23],
Y = mass^k_1_^ · height^k_2_^ · exp(a + b · MO + c · MO^2^) · ε.(1)

This model has the advantages of having proportional body size components and the flexibility of maturity-offset estimates (entered as a quadratic) within an exponential term that will ensure that the measure of physical performance (Y) will always remain non-negative irrespective of the subjects’ maturity-offset estimates. Note that “ε”, the multiplicative error ratio, also assumes the error will increase in proportion to the physical performance variable Y. This evidence of heteroscedasticity can be clearly seen in Figure 1.

The model (Equation (1)) can be linearized with a log transformation (ln = log_e_). A linear regression analysis on ln(Y) can then be used to estimate the unknown parameters of the log transformed model:ln(Y) = k_1_·ln(height) + k_2_·ln(mass) + a + b · MO + c · MO^2^ + ln(ε).(2)

Further categorical or group differences within the population, e.g., sex, age (entered as discrete categories, 8 to 15 yrs.), can be explored by allowing the constant intercept “a” parameter or slope “b” and “c” parameters in Equation (2) to vary for each group (particularly sex) within the ANCOVA (note that the terms ln(height), ln(mass), MO, and MO2 in Equation (2) are the covariates). The significance level was set at *p* < 0.05.

Inter-rater reliability was assessed using the ICC and the SEM. The significance level was set at *p* < 0.05. The statistical analyses were conducted using the statistical software SPSS version 26.

## 3. Results

The number of Tunisian children together with the mean ± standard deviation body mass, height and maturation offset by age group are given in Table 1.

### 3.1. Sprint Speed

The multiplicative model relating the 10, 20 and 30 m sprint speeds (m s^−1^) as the dependent variables (DP) to the body size and maturation offset characteristics are given in Table 2.

The constant “a” varied significantly with the main effects of sex and age (*p* < 0.001), but there was no interaction between age and sex.

The body mass and height exponents (Table 2) associated with the three sprint speeds for the above models can be rearranged and expressed as a height-to-mass ratio within a curvilinear power function as follows:Height-to-mass ratio (Speed 30 m) = mass^−0.16^·height^0.71^ = (height ·mass*^−^*^0.23^) ^0.71^,
Height-to-mass ratio (Speed 20 m) = mass^−0.15^·height^0.45^ = (height ·mass*^−^*^0.32^) ^0.45^,
Height-to-mass ratio (Speed 10 m) = mass^−0.14^·height^0.34^ = (height ·mass*^−^*^0.42^) ^0.34^,
since, for example, the 20 m Speed model mass^−0.15^ = (mass^−0.32^)^0.45^. The 95% confidence interval (CI) for the rearranged/rescaled mass exponent −0.32 is (−0.39 to −0.26). Note that this height-to-body mass ratio is similar to the reciprocal Ponderal index (RPI) = height ·mass*^−^*^0.333^, since the 95% CI’s encompasses −0.333.

### 3.2. Horizontal Jumping

The multiplicative model relating the log-transformed SLJ and 5JT (m) as the dependent variables (DP) to the body size and maturation offset characteristics are given in Table 3.

The constant “a” varied significantly for the log-transformed SLJ with the main effects of sex and age plus their interaction (all *p* < 0.001). For the log-transformed 5JT, significant differences were identified for the age and sex main effects (*p* < 0.001) but their interaction (*p* = 0.049). These interactions are shown in Figure 2a,b, respectively, below.

The body mass and height exponents associated with the SLJ (Table 3) and 5JT (Table 3) can be rearranged and expressed as a height-to-mass ratio within a curvilinear power function as follows:Height-to-mass ratio (LnSLJ) = mass^−0.25^·height^1.45^ = (height ·mass*^−^*^0.17^)^1.45^,
Height-to-mass ratio (LnFIVE) = mass^−0.16^·height^1.26^ = (height ·mass*^−^*^0.12^)^1.26^,
since, for example, in the LnSLJ model mass^−0.25^ = (mass^−0.17^)^1.45^. Note that both these height-to-body mass ratios could be described as “extreme” RPI, where the mass exponents are considerably less (mass*^−^*^0.17^ and mass*^−^*^0.12^, respectively) than the anticipated mass exponent (mass*^−^*^0.333^) for the RPI.

### 3.3. Vertical Jumping

The multiplicative model relating the log-transformed CMJ and SJ tests (cm) as the dependent variables (DP) to the body size and maturation offset characteristics are given in Table 4.

The sex and age group main effects together with their interaction varied significantly for log-transformed CMJ test (*p* < 0.001, *p* = 0.002, and 0.008, respectively). For the log-transformed SJ test, significant differences were identified for the age and sex main effects (*p* < 0.001), but there was no significant interaction effect (*p* > 0.05). The lack of the age-by-sex interaction can be explained better by the sex-by-MO and sex-by-MO^2^ interactions reported in Table 4 and plotted in Figure 3.

Both interactions can be seen in Figure 4a,b respectively below.

The body mass and height exponents associated with the CMJ and SJ tests (Table 4) can be rearranged and expressed as a height-to-mass ratio within a curvilinear power function as follows:Height-to-mass ratio (LnCMJ) = mass^−0.47^·height^1.43^ = (height ·mass*^−^*^0.33^)^1.43^,
Height-to-mass ratio (LnSJ) = mass^−0.51^·height^1.70^ = (height ·mass*^−^*^0.30^) ^1.70^,
since, for example, in the LnCMJ model mass^−0.47^ = (mass^−0.33^)^1.43^. Note that both these height-to-body mass ratios are approximately the RPI, where the mass exponents mass*^−^*^0.33^ and mass*^−^*^0.30^, respectively, are very similar to the anticipated mass exponent (mass*^−^*^0.333^) for the RPI.

### 3.4. Strength and Change of Direction Tests

The multiplicative model relating the log-transformed handgrip strength test (kg) and ICoDT speed (m.s^−1^) as the dependent variables (DP) to the body size and maturation offset characteristics are given in Table 5.

The significant quadratic association between hand grip strength (kg) and MO (yrs) reported in Table 5 can be seen in Figure 1.

The age group and sex main effects for both log-transformed handgrip strength and the ICoDT were significant (all *p* < 0.001) with significant age-by-sex interactions (*p* = 0.045 and *p* = 0.004, respectively). These interactions are given in Figure 5a,b, respectively, below.

For the log-transformed handgrip strength, both the body mass and height exponents are positive (Table 5) indicating that being taller and heavier are associated with better handgrip strength performances. As for the log-transformed ICoDT speeds, the height and mass exponents reported in Table 5 can be rearranged and expressed as a height-to-mass ratio within a curvilinear power function as follows:

Height-to-mass ratio (LnICoDT) = mass^−0.11^·height^0.21^ = (height ·mass*^−^*^0.51^)^0.21^, since the mass^−0.11^ = (mass^−0.51^)^0.21^. Note that this height-to-body mass ratio is approximately the inverted body mass index (iBMI = height ·mass*^−^*^0.5^).

## 4. Discussion

The present study used allometric modelling to identify optimal body size and shape characteristics (using height, body mass and height-to-weight ratios) associated with nine PF tests of Tunisian children and adolescents. The model incorporated other developmental factors and confounders including MO, sex, and chronological age, all likely to be associated with PF performance. While prior research has examined the effect of maturation in a similar way to the present study [23], maturation was not introduced as a continuous quadratic. The present study is the first to consider that MO is curvilinear in nature. The study identified just how important maturity (offset) is at predicting the boys’ and girls’ PF performance tests during childhood and adolescence, having also controlled for differences body size, shape, sex and chronological age. It is now clear that boys who go through peak height velocity (PHV) at an earlier/younger age (with a more positive MO score) will perform better than boys who go through peak height velocity at a later/older age (who have negative MO scores), assuming the same body size/shape and chronological age. The positive linear and quadratic MO terms in Table 2, Table 3, Table 4 and Table 5 confirm this association for all boy’s fitness test performances although the MO^2^ term was significant in 7/9 tests, the two exceptions being the 5JT and the ICoDT that arguably require a greater element of technical skill rather than physical prowess. This is supported by previous research which suggests that motor competence may initially decrease after the occurrence of PHV [24].

In contrast, the majority of the girl’s PF tests peak at approximately peak PHV, and decline thereafter identified by the negative female quadratic MO^2^ terms (Table 2, Table 3, Table 4 and Table 5 and Figure 3). This suggest that girls who go through their PHV either too early (very positive maturity-offset scores) or too late in relation to their chronological age (very negative maturity-offset scores) will be disadvantaged (perform less well at most PF tests). Girls whose MO are near the middle of the range, would appear to have a distinct performance test advantage, (Table 2, Table 3, Table 4 and Table 5 and Figure 3). The chronological age associated with a female maturity-offset value of zero was estimated to be 13.5 years.

The allometric models also identified the optimal body size and shape characteristics associated with the nine PF tests. All nine models identified body mass as the key (most important) predictor variable of the PF tests, eight being negative, the only exception being handgrip strength that had a positive mass exponent (see the t scores in Table 2, Table 3, Table 4 and Table 5). The second most important predictor was invariably height that was found to be positive in all 9 models. Taken together, these results suggest that the optimal body size and shape associated with success in PF tests was to be tall and light, which can be expressed as having a greater height-to-weight ratio. These findings are in line with earlier studies. For example, Lovecchio et al. [25] identified the optimal body-shape and composition associated with PF levels of children living in urban and rural areas of Italy. The same authors revealed that taller and more linear or ectomorphic children perform better on measures of PF tests (i.e., standing broad jump test, sit-up test). Likewise, Nevill et al. [22] suggested that the inverse ponderal index is the most appropriate body-shape indicator associated with running and jumping activities among Greek children. In addition, Bustamante et al. [23] showed that RPI was found to be the most suitable body-shape indicator associated with the standing long jump, handgrip, and the shuttle run speed tests among Brazilian and Peruvian schoolchildren. In agreement with our findings, the same authors revealed that children who go through PHV at an earlier age (i.e., more positive MO score) displayed better PF performance.

However, the nature of these ratios varied considerably depending on which test. In three of the tests including the two vertical jump test (CMJ and SJ) and 20 m sprint speed, the reciprocal Ponderal index (RPI = height ·mass*^−^*^0.333^) was the ideal body shape associated with success. The optimal shape associated with the two horizontal jump tests (SLJ and five-jump) test plus 30-m sprint speed would appear to be an even more “extreme” linear RPI shape (approximately = height ·mass*^−^*^0.20^). The optimal body shape associated with 10-m sprint speed and the ICoDT test was identified as the inverse BMI (height.mass^−0.5^), a ratio thought to reflect leanness rather than excess body mass [26,27].

The “extreme” linear RPI shape associated with the two horizontal jumps confirms the need for being more angular together with a less important “weighting” associated with body mass (mass*^−^*^0.20^). In contrast, the shape associated with the two vertical jump tests, body mass, is likely to have a greater detrimental effect, hence the great negative body mass term in the ratio (RPI = height · mass*^−^*^0.333^). Unsurprisingly the model for handgrip strength identified both the height and body mass exponents to be positive, suggesting that being taller (having greater leverage) and having greater body mass (including muscle mass) will benefit handgrip strength performance.

As with all studies, the findings need to be considered in light of some limitations. Firstly, the children in the present study were all recruited from Tunisia and so generalizations to other populations have to be made with caution. In addition, in relation to PHV, previous research has found intra-individual variation in predicted age at PHV to be large, and therefore relatively few predicted ages at PHV in early and late maturing children approximate observed age at PHV [28]. Nonetheless, given decisions made about PHV in youth performance contexts also use the equations we have or very similar equations with the same limitations, our findings remain valid in these contexts.

The findings of the present study have important implications for talent scouts and coaches wishing to recruit children into their sports/athletic clubs. Coaches searching for potentially successful youths should ideally recruit relatively tall and light children (with high or “extreme” RPIs) for sporting activities that involve running and jumping. However, given boys of a similar body size, shape and chronological age (i.e., all other things equal), scouts and coaches would be wise to recruit early rather than late maturers to further optimise the boys’ physical performance potential. However, this strategy is not true for girls. Similar to boys, girls who are relatively tall and light are once again the most likely shape to succeed at dynamic activities/sports that involve running and jumping. However, all other things being equal, late and early maturers are more likely to be disadvantaged at such physical activities compared with girls who mature in the middle of this maturity age range.

## 5. Conclusions

The current study has confirmed that being tall and light is the ideal shape to be successful at most PF test, but the height-to-weight ratio varied considerably from test to test. For example, in the vertical jump tests, the reciprocal Ponderal index (RPI = height ·mass*^−^*^0.333^) was the ideal body shape associated with success. For horizontal jump tests, where being taller was key (mass had a little detrimental effect), a more “extreme” linear RPI shape (approximately = height ·mass*^−^*^0.20^) appears to be a more suitable shape. Having controlled for body size/shape, sex, and chronological age, the model also identified MO as an additional significant predictor. Boys who go through PHV at an earlier/younger age will outperform their peers who go through PHV at a later/older age (Figure 3). In contrast, most of the girls’ PF tests peak at approximately PHV and decline thereafter. This suggests that girls whose MO is near the middle of the age range would appear to have a distinct advantage at the PF tests compared to girls who are either early or late maturers.

## Figures and Tables

**Figure 1 ijerph-18-00497-f001:**
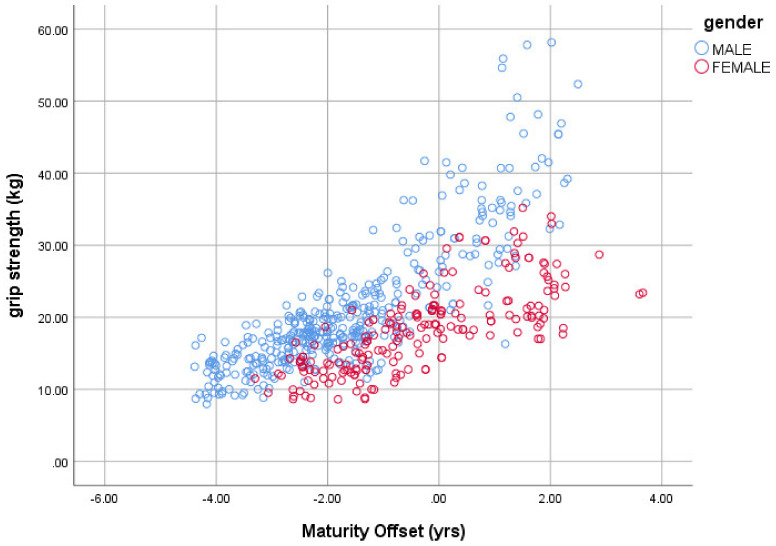
The association between hand grip strength (kg) and maturity offset (years).

**Figure 2 ijerph-18-00497-f002:**
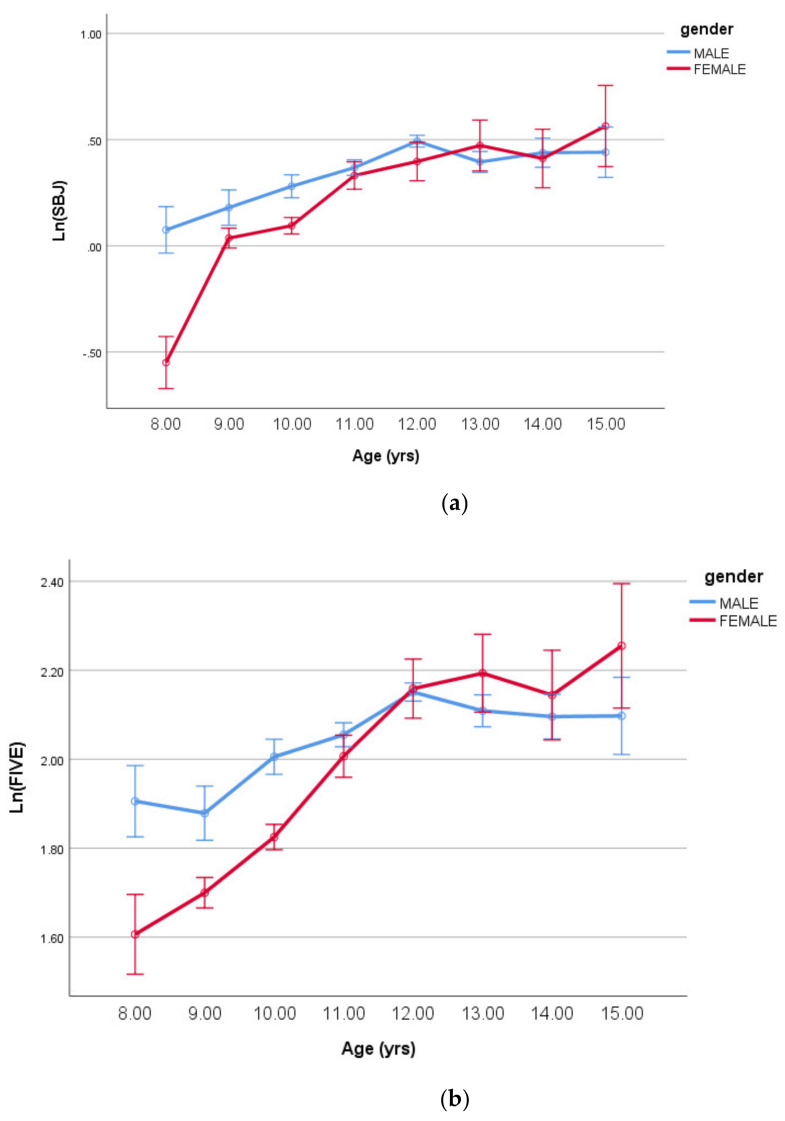
(**a**) Log-transformed standing long jump (means ± SE) by age having controlled for body size/shape and maturity offset (covariates in Table 3). (**b**) Log-transformed five jump test (means ± SE) by age having controlled for body size/shape and maturity offset (covariates in Table 3).

**Figure 3 ijerph-18-00497-f003:**
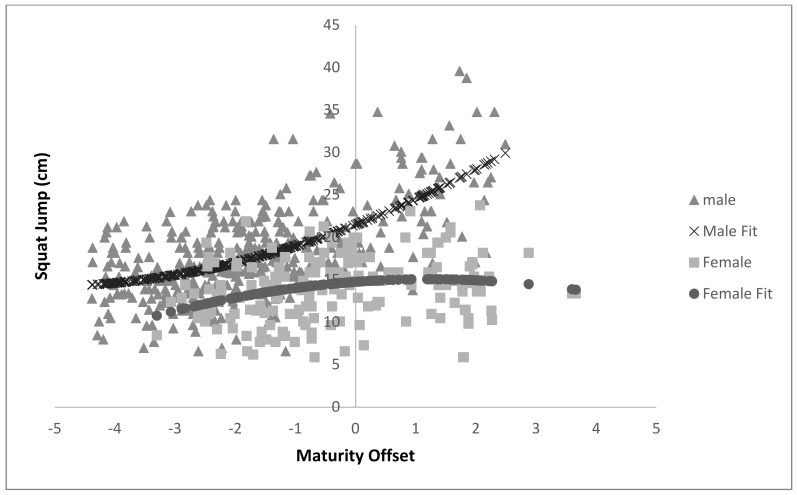
The quadratic associations between squat jump (cm) and maturity offset (yrs) for boys and girls.

**Figure 4 ijerph-18-00497-f004:**
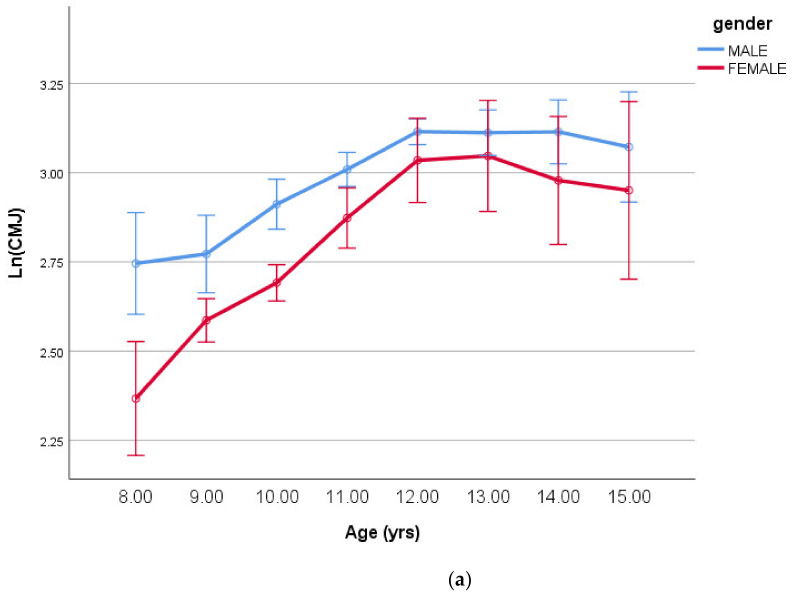
(**a**) Log-transformed counter-movement jump (means ± SE) by age having controlled for body size/shape and maturity offset (covariates in Table 4). (**b**) Log-transformed squat jump (means ± SE) by age having controlled for body size/shape and maturity offset (covariates in Table 4).

**Figure 5 ijerph-18-00497-f005:**
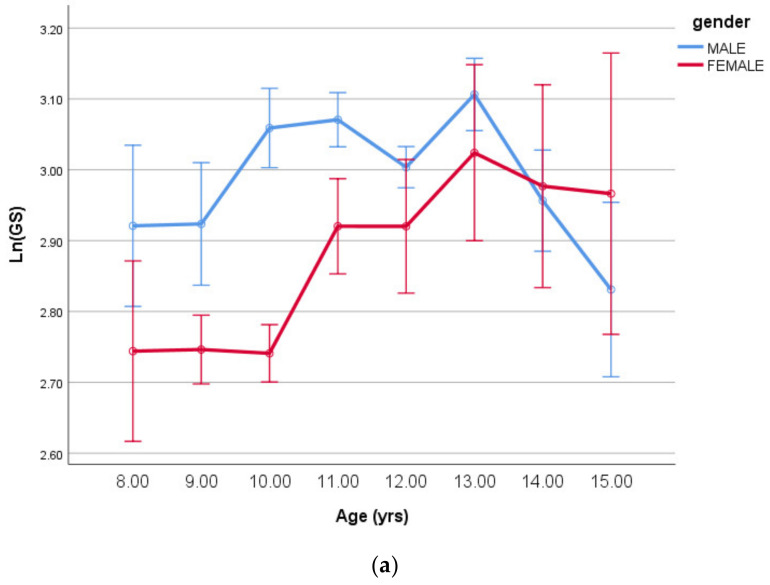
(**a**) Log-transformed handgrip strength (means ± SE) by age having controlled for body size/shape and maturity offset (covariates in Table 5). (**b**) Log-transformed Illinois Change of direction test (means ± SE) by age having controlled for body size/shape and maturity offset (covariates in Table 5 DV LnICoDT).

**Table 1 ijerph-18-00497-t001:** The number of children in the sample, plus the mean (±SD) body mass, height and maturation offset by sex and age group.

Boys									
Age	N	Body Mass (kg)	SD	Height (cm)	SD	BMI (kg·m^−2^)	SD	MO (Years)	SD
8	45	30.64	5.6	132.01	6.34	17.49	2.26	−3.93	0.27
9	36	37.06	9.3	140.88	8.05	18.56	3.81	−3.16	0.35
10	85	37.94	8.68	143.34	5.87	18.38	3.54	−2.53	0.31
11	75	42.24	9.65	148.31	7.12	19.07	3.39	−1.85	0.35
12	74	45.86	10.06	154.64	8.73	19.05	3.25	−1.03	0.44
13	22	49.78	15.86	161.18	9.41	18.86	4.36	−0.11	0.44
14	44	59.66	13.87	172.01	7.36	20.01	3.84	0.96	0.42
15	15	68.84	14.46	177.61	4.91	21.78	4.46	1.97	0.34
Total	396	43.56	13.83	150.36	14.17	18.89	3.56	−1.64	1.6
**Girls**									
8	3	29.3	3.3	128.97	4.91	17.57	0.66	−3.02	0.32
9	32	31.18	7.79	135.48	7.46	16.88	3.13	−2.30	0.35
10	43	41.06	10.36	142.96	5.84	19.93	4.08	−1.38	0.33
11	50	46.26	12.53	150.75	6.83	20.17	4.26	−0.37	0.37
12	27	53.54	13.01	155.5	7.09	22.15	5.42	0.49	0.46
13	17	56.51	13.24	160.09	5.27	22.01	4.83	1.44	0.42
14	24	52.97	16.74	157.5	2.59	21.39	6.87	1.86	0.19
15	5	52.74	4.61	166	9.59	19.26	2.38	2.93	0.68
Total	201	45.3	14.4	148.94	10.71	20.1	4.85	−0.32	1.51

MO = Maturity offset; SD = Standard deviation.

**Table 2 ijerph-18-00497-t002:** The fitted ANCOVA parameters predicting log-transformed 30, 20 and 10 m speed (m.s^−1^) using the covariates described in Equation (2).

Parameter Estimates						
DP = LnSpeed30	B	SE	t	*p*	Lower Bound	Upper Bound
Intercept	−1.255	0.825	−1.522	0.129	−2.875	0.365
Female	0.119	0.084	1.406	0.160	−0.047	0.285
LnMass	−0.158	0.017	−9.376	0.000	−0.192	−0.125
LnHT	0.707	0.169	4.192	0.000	0.376	1.039
MO	0.031	0.022	1.413	0.158	−0.012	0.073
MO^2^	0.009	0.004	2.672	0.008	0.003	0.016
Female * MO	−0.069	0.020	−3.532	0.000	−0.107	−0.031
Female * MO^2^	−0.017	0.006	−2.801	0.005	−0.029	−0.005
The coefficient of determination R^2^ = 0.592 (Adj R^2^ = 0.577); log transformed error ratio being 0.073 or 7.6%, having taken antilogs						
DP = LnSpeed20						
Intercept	−0.083	0.698	−0.118	0.906	−1.454	1.289
Female	0.122	0.071	1.716	0.087	−0.018	0.262
LnMass	−0.145	0.014	−10.170	0.000	−0.173	−0.117
LnHT	0.452	0.143	3.166	0.002	0.172	0.733
MO	0.046	0.018	2.489	0.013	0.010	0.082
MO^2^	0.008	0.003	2.533	0.012	0.002	0.013
Female * MO	−0.065	0.016	−3.934	0.000	−0.097	−0.032
Female * MO^2^	−0.018	0.005	−3.494	0.001	−0.028	−0.008
The coefficient of determination R^2^ = 0.605 (Adj R^2^ = 0.591); log transformed error ratio being 0.061 or 6.3%, having taken antilogs						
DP = LnSpeed10						
Intercept	0.362	0.698	0.519	0.604	−1.009	1.733
Female	0.100	0.071	1.400	0.162	−0.040	0.240
LnMass	−0.144	0.014	−10.041	0.000	−0.172	−0.116
LnHT	0.340	0.143	2.379	0.018	0.059	0.621
MO	0.045	0.018	2.460	0.014	0.009	0.081
MO^2^	0.003	0.003	1.125	0.261	−0.003	0.009
Female * MO	−0.066	0.016	−3.990	0.000	−0.098	−0.033
Female * MO^2^	−0.009	0.005	−1.845	0.066	−0.019	0.001

The coefficient of determination R^2^ = 0.547 (Adj R^2^ = 0.530); log transformed error ratio being 0.061 or 6.3%, having taken antilogs. * = product.

**Table 3 ijerph-18-00497-t003:** The fitted ANCOVA parameters predicting log-transformed standing long jump Ln (SLJ) and the five jump test Ln (FIVE) using the covariates described in Equation (2).

Parameter Estimates						
DP = LnSLJ	B	SE	t	*p*	Lower Bound	Upper Bound
Intercept	−5.944	1.847	−3.218	0.001	−9.571	−2.316
Female	0.082	0.189	0.434	0.665	−0.289	0.453
LnMass	−0.249	0.038	−6.582	0.000	−0.323	−0.174
LnHT	1.451	0.378	3.839	0.000	0.709	2.193
MO	0.019	0.049	0.381	0.703	−0.077	0.114
MO^2^	0.017	0.008	2.137	0.033	0.001	0.033
Female * MO	−0.107	0.044	−2.437	0.015	−0.193	−0.021
Female * MO^2^	−0.020	0.013	−1.501	0.134	−0.046	0.006
The coefficient of determination R^2^ = 0.527 (Adj R^2^ = 0.510); log transformed error ratio being 0.163 or 17.7%, having taken antilogs						
DP = LnFIVE						
Intercept	−3.645	1.352	−2.695	0.007	−6.301	−0.989
Female	0.064	0.138	0.461	0.645	−0.208	0.336
LnMass	−0.155	0.028	−5.598	0.000	−0.209	−0.100
LnHT	1.261	0.277	4.558	0.000	0.718	1.805
MO	0.028	0.036	0.798	0.425	−0.042	0.098
MO^2^	0.010	0.006	1.669	0.096	−0.002	0.021
Female * MO	−0.103	0.032	−3.200	0.001	−0.166	−0.040
Female * MO^2^	−0.007	0.010	−0.692	0.489	−0.026	0.012

The coefficient of determination R^2^ = 0.661 (Adj R^2^ = 0.650); log transformed error ratio being 0.119 or 12.6%, having taken antilogs. * = product.

**Table 4 ijerph-18-00497-t004:** The fitted ANCOVA parameters predicting log-transformed countermovement jump Ln(CMJ) and the squat jump tests Ln(SJ) using the covariates described in Equation (2).

Parameter Estimates						
DP = LnCMJ	B	SE	t	*p*	Lower Bound	Upper Bound
Intercept	−2.349	2.406	−0.976	0.329	−7.075	2.377
Female	−0.205	0.246	−0.833	0.405	−0.689	0.279
LnMass	−0.466	0.049	−9.478	0.000	−0.563	−0.370
LnHT	1.429	0.492	2.901	0.004	0.462	2.396
MO	0.090	0.063	1.427	0.154	−0.034	0.215
MO^2^	0.029	0.010	2.764	0.006	0.008	0.049
Female * MO	−0.155	0.057	−2.708	0.007	−0.267	−0.042
Female * MO^2^	−0.024	0.017	−1.351	0.177	−0.058	0.011
The coefficient of determination R^2^ = 0.487 (Adj R^2^ = 0.468) with the log transformed error ratio being 0.212 or 23.7%, having taken antilogs						
DP = LnSJ						
Intercept	−3.706	2.677	−1.384	0.167	−8.965	1.553
Female	−0.084	0.274	−0.305	0.761	−0.622	0.455
LnMass	−0.505	0.055	−9.225	0.000	−0.613	−0.398
LnHT	1.702	0.548	3.107	0.002	0.626	2.778
MO	0.081	0.070	1.149	0.251	−0.057	0.219
MO^2^	0.035	0.012	3.027	0.003	0.012	0.057
Female * MO	−0.165	0.064	−2.600	0.010	−0.290	−0.040
Female * MO^2^	−0.045	0.019	−2.307	0.021	−0.083	−0.007

The coefficient of determination R^2^ = 0.475 (Adj R^2^ = 0.456) with the log transformed error ratio being 0.236 or 26.7%, having taken antilogs. * = product.

**Table 5 ijerph-18-00497-t005:** The fitted ANCOVA parameters predicting log-transformed grip strength Ln(GS) and the Illinois Change of direction test Ln(ICoDT) using the covariates described in Equation (2).

Parameter Estimates						
DV = LnGS	B	SE	T	*p*	Lower Bound	Upper Bound
Intercept	−4.790	1.920	−2.495	0.013	−8.560	−1.019
Female	0.036	0.197	0.186	0.853	−0.350	0.423
LnMass	0.378	0.039	9.617	0.000	0.300	0.455
LnHT	1.254	0.393	3.193	0.001	0.483	2.026
MO	0.149	0.050	2.948	0.003	0.050	0.248
MO^2^	0.025	0.008	3.025	0.003	0.009	0.041
Female * MO	−0.147	0.046	−3.235	0.001	−0.237	−0.058
Female * MO^2^	−0.018	0.014	−1.294	0.196	−0.045	0.009
The coefficient of determination, R^2^ = 0.793 (Adj R^2^ = 0.786) with the log transformed error ratio being 0.169 or 18.5%, having taken antilogs						
DV = LnICoDT						
Intercept	0.489	0.871	0.561	0.575	−1.223	2.200
Female	0.023	0.089	0.263	0.792	−0.152	0.199
LnMass	−0.107	0.018	−5.984	0.000	−0.142	−0.072
LnHT	0.207	0.178	1.162	0.246	−0.143	0.558
MO	0.037	0.023	1.616	0.107	−0.008	0.082
MO^2^	0.002	0.004	0.642	0.521	−0.005	0.010
Female * MO	−0.024	0.021	−1.155	0.249	−0.064	0.017
Female * MO^2^	−0.013	0.006	−2.071	0.039	−0.026	−0.001

The coefficient of determination, R^2^= 0.630 (Adj R^2^ = 0.616) with the log transformed error ratio being 0.077 or 8.0%, having taken antilogs. * = product.

## Data Availability

The data presented in this study are available on request from the corresponding author.

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
