# Peer review of "Are Early or Late Maturers Likely to Be Fitter in the General Population?"

_ijerph, 2021, doi:10.3390/ijerph18020497_

Round 1
Reviewer 1 Report
General comment
The authors present a study that aimed to examine the association between biological maturation and physical fitness as a continuous process in a sample of non-specifically trained schoolchildren. The study appears well designed and structured. I have highlighted some topics that would benefit from more detail and that are list below on the Specific Comments. Most of them are minor corrections that are meant to improve the readability of the manuscript; however, some of the authors' reasoning needs to be explained further, as well as my concerns.
Specific comments
Abstract - Line 16 should be revised. Please, include the following: “The present study aims to…”.
Introduction - This section is clear and well developed and provides a good rationale for the study. Add the abbreviation of physical fitness in line 33; i.e. “The assessment of physical fitness (PF)…”. Please, revise the lines 37 and 38 with the above mention.
Materials and Methods:
- Did you calculate the minimal sample size? Please, include the procedures that were considered to choose 597 subjects.
- The age of subjects ranged from 8 to 15 years-old. So, the concept of “children” and “adolescents” may need to be clarified.
- Inclusion and exclusion criteria were considered? If yes, I think that authors should include in “subjects” subsection. You should include additional information about the subjects. Were they a random sample or recruited specifically for this reason?
- The recruited sample consisted of boys and girls who were not engaged in specific training. However, how was it controlled? What do you mean with “non-specifically trained schoolchildren”?
Procedures. It is not clear which variables were assessed on the anthropometric measurements, as well as the instruments used for the assessment.
Statistical Analysis. Please, clarify the statistical tests used in the present study (besides the model). At some point, it gets slightly jumble. Additional, clarify the use of ANOVA and/or ANCOVA tests.
Results. The sample size of each group, considering the genders, may affect the data. For example, 45 boys were analysed in the age-group of 8 years-old, whereas only 3 girls were assessed in the same age-group. Could you please comment and justify on that concern?
Discussion:
- The discussion is limited to the obvious findings with an exhaustive description. The authors do not support findings with available literature.
- What is the novelty of the findings and their practical implications (according to the results obtained in this study)? Please, revise this section after analysing the previous comments, as well as the limitations of the present study.
Author Response
Dear reviewer,
We thank you for your valuable time and for the constructive and helpful comments. We carefully addressed all of your concerns and suggestions in the following point-by-point statement. Amendments to the manuscript were made whenever necessary.
Best Regards,
Dr. Helmi Chaabene (on behalf of all authors)
Reviewer 1 (changes were highlighted in YELLOW)
General comment
The authors present a study that aimed to examine the association between biological maturation and physical fitness as a continuous process in a sample of non-specifically trained schoolchildren. The study appears well designed and structured. I have highlighted some topics that would benefit from more detail and that are list below on the Specific Comments. Most of them are minor corrections that are meant to improve the readability of the manuscript; however, some of the authors' reasoning needs to be explained further, as well as my concerns.
Specific comments
Comment 1
Abstract - Line 16 should be revised. Please, include the following: “The present study aims to…”
Authors’ response to comment 1: Added as suggested. Thank you.
Comment 2:
Introduction - This section is clear and well developed and provides a good rationale for the study. Add the abbreviation of physical fitness in line 33; i.e. “The assessment of physical fitness (PF)…”. Please, revise the lines 37 and 38 with the above mention.
Authors’ response to comment 2: Thank you for your affirmative comment. The abbreviation of “physical fitness” was introduced at first appearance and used consistently throughout the entire manuscript.
Materials and Methods:
Comment 3:
- Did you calculate the minimal sample size? Please, include the procedures that were considered to choose 597 subjects.
Authors’ response to comment 3: Thank you for this comment. Sample size was not calculated a priori. Please note that our sample was a “purposive” sample (Guarte et al., 2006). Specifically, a purposive sample is a non-probability sample that is selected based on characteristics of a population and the objective of the study. Purposive sampling is different from convenience sampling and is also known as judgmental, selective, or subjective sampling (Guarte et al., 2006). We chose the sample thought to be a fair representation of children from an urban area in Tunisia.
Comment 4: - The age of subjects ranged from 8 to 15 years-old. So, the concept of “children” and “adolescents” may need to be clarified.
Authors’ response to comment 4: In accordance with the reviewer’s comment, this issue has now been addressed in the revised manuscript.
“In accordance with Lloyd et al. (2014), we defined the following sex specific age ranges for children and adolescents. The term ‘children’ refers to girls and boys, generally up to the age of 11 and 13 years, respectively. The term ‘adolescence’ refers to girls 12–18 years and boys 14–18 years.”
Comment 5:
- Inclusion and exclusion criteria were considered? If yes, I think that authors should include in “subjects” subsection. You should include additional information about the subjects. Were they a random sample or recruited specifically for this reason?
Authors’ response to comment 5: Thank you for your comment. We added more details relative to the recruited sample as well as exclusion/inclusion criteria.
“To be enrolled in this study, participants had to be apparently healthy (no diagnosed disease or injury) and not engaged in any structured training program. In this sense, children who were members of sport clubs were excluded from study participation. All schoolchildren were from families of low-to-medium socioeconomic status and enrolled in public elementary or high schools in an urban area of Tunisia.”
Comment 6:
- The recruited sample consisted of boys and girls who were not engaged in specific training. However, how was it controlled? What do you mean with “non-specifically trained schoolchildren”?
Authors’ response to comment 6: Thank you for the opportunity to clarify this point. All recruited schoolchildren were not engaged in any form of structured training (e.g., not enrolled in sports clubs) for at least 6 months before the start of the study. The following statement was added to the revised version of the manuscript:
“In this sense, children who were members of sport clubs were excluded from study participation.”
Comment 7:
It is not clear which variables were assessed on the anthropometric measurements, as well as the instruments used for the assessment.
Authors’ response to comment 7: Thank you for this comment. We have clarified the issues raised by the reviewer in the respective paragraphs (Methods section):
“Each individual’s height (m) and body-mass (kg) were assessed to the nearest 0.1 cm and 0.1 kg, using a SECA stadiometer and a SECA weighing scale (SECA Instruments Ltd, Hamburg, Germany). Body mass index (BMI) was calculated using body-mass divided by height squared (kg/m2). All anthropometric measurements were recorded twice by two raters and the mean scores were retained for further statistical analysis. Inter-rater reliability was assessed using intraclass correlation coefficients (ICC) and standard error of measurement (SEM). For ICCs values were > 0.90 and for SEM < 5%.
Besides chronological age, biological maturity was estimated using the maturity offset (MO) method. For both sexes, MO was calculated by predicting age at peak-height-velocity using the predictive equations established by Moore et al. [16].
For girls: MO= −7.709133 + (0.0042232 x age height).
For boys: MO= ”
Comment 8:
Please, clarify the statistical tests used in the present study (besides the model). At some point, it gets slightly jumble. Additional, clarify the use of ANOVA and/or ANCOVA tests.
Authors’ response to comment 8: Apologies for the confusion, we are always using ANCOVA. This point together with more details on the analysis have now been added to the revised manuscript:
“Further categorical or group differences within the population, e.g. sex, age (entered as discrete categories, 8 to 15 yrs.), can be explored by allowing the constant intercept ‘a’ parameter or slope ‘b’ and ‘c’ parameters in Eq. 2 to vary for each group (particularly sex) within the ANCOVA (note that the terms ln(height), ln(mass), MO, and MO2 in Eq. 2 are the covariates). The significance level was set at p<0.05.”
Comment 9:
The sample size of each group, considering the genders, may affect the data. For example, 45 boys were analysed in the age-group of 8 years-old, whereas only 3 girls were assessed in the same age-group. Could you please comment and justify on that concern?
Authors’ response to comment 9: We agree with the reviewer that the respective sample sizes vary within the age groups. However, the error bars in all figures will accurately reflect these differences leaving the reader in no doubt where such differences are important. Additionally, we allowed each child who adheres to the purposive sampling criteria to participate as long as he/she wanted to. This has resulted in a different number of males and females per age category. However, we think that it is unethical to exclude a child who complies with the inclusion criteria for the sake of homogeneity.
Comment 10:
- The discussion is limited to the obvious findings with an exhaustive description. The authors do not support findings with available literature.
Authors’ response to comment 10: In accordance with the reviewer’s comment, we have included more studies and discussed their findings in the context of our own study results.
“The allometric models also identified the optimal body size and shape characteristics associated with the nine PF tests. All nine models identified body mass as the key (most important) predictor variable of the PF tests, eight being negative, the only exception being handgrip strength that had a positive mass exponent (see the t scores in tables 2-5). The second most important predictor was invariably height that was found to be positive in all 9 models. Taken together, these results suggest that the optimal body size and shape associated with success in PF tests was to be tall and light, which can be expressed as having a greater height-to-weight ratio. These findings are in line with earlier studies. For example, Lovecchio et al. [25] identified the optimal body-shape and composition associated with PF levels of children living in urban and rural areas of Italy. The same authors revealed that taller, and more linear or ectomorphic children perform better on measures of PF tests (i.e., standing broad jump test, sit-up test). Likewise, Nevill et al. [22] suggested that the inverse ponderal index is the most appropriate body-shape indicator associated with running and jumping activities among Greek children. In addition, Bustamante et al. [23] showed that RPI was found to be the most suitable body-shape indicator associated with the standing long jump, handgrip, and the shuttle run speed tests among Brazilian and Peruvian schoolchildren. In agreement with our findings, the same authors revealed that children who go through PHV at an earlier age (i.e., more positive MO score) displayed better PF performance.”
Comment 11:
- What is the novelty of the findings and their practical implications (according to the results obtained in this study)? Please, revise this section after analyzing the previous comments, as well as the limitations of the present study.
Authors’ response to comment 11: Thank you for this comment. Our findings are novel in as much as they showed that being tall and light is the ideal shape to be successful at most physical fitness tests. Having controlled for body-size/shape, sex, and chronological age, the model identified maturity-offset as an additional predictor. Boys who go earlier/younger through peak-height-velocity (PHV) outperform those who go at a later/older age. However, most of the girls’ physical-fitness tests peaked at PHV age and declined thereafter. Girls whose age at PHV was near the middle of the age range would appear to have an advantage compared to early or late maturers. These findings have important implications for talent scouts and coaches wishing to recruit children into their sports/athletic clubs. Additionally, the important aspect of the present study, which is often overlooked in prior work, is the focus here on untrained children. The majority of prior studies on this topic have focused on children in talent development programmes who are already specializing in a sport. There is novelty, in our opinion, in identifying how maturation might influence fitness performance in the general untrained population of children. This in a sense provides a cleaner baseline of the effects of maturation on children compared to studies embedded in sport performance. Moreover, most of the earlier studies have examined differences in outcome variables across different maturation stages. Of note, such approaches ignore the fact that the effect of maturation on physical fitness outcome variables is likely to be a continuous, dynamic and non-linear process (Towlson et al., 2018).
References used during the revision
- Lloyd, R.; Faigenbaum, A.; Stone, M.; Oliver, J.; Jeffreys, I.; Moody, J.; Brewer, C.; Pierce, K.; McCambridge, T.; Howard, R.; Herrington, L.; Hainline, B.; Micheli, L.; Jaques, R.; Kraemer, W.; McBride, M.; Best, T.; Chu, D.; Alvar, B.; Myer, G. Position statement on youth resistance training: the 2014 International Consensus. Br J Sports Med. 2014; 48(7), 498-505.
- Guarte, Jacqueline M., and Erniel B. Barrios. "Estimation under purposive sampling." Communications in Statistics-Simulation and Computation 35.2 (2006): 277-284.
- Nevill, A.; Tsiotra, G.; Tsimeas, P.; Koutedakis, Y. Allometric associations between body-size, shape and physical performance of Greek children. Pediatr Exerc Sci. 2009, 21(2) 220-232, 2009.
- Bustamante Valdivia, A., Maia, J., & Nevill, A. (2015). Identifying the ideal body size and shape characteristics associated with children’s physical performance tests in Peru. Scandinavian Journal of Medicine & Science in Sports, 25(2), e155–6.
- Lovecchio, N.; Giuriato,M.: Zago, M.; Nevill, A.M. Identifying the optimal body shape and composition associated with strength outcomes in children and adolescent according to place of residence: An allometric approach. J Sports Sci. 2019; 37(12),1434-1441
- Towlson, C.; Cobley, S.; Parkin, G.; Lovell, R. When does the influence of maturation on anthropometric and physical fitness characteristics increase and subside? Scand J Med Sci Sports.2018; 28: 1946– 1955.
- Lloyd, R.S.; Oliver, J.L.; Radnor, J.M.; Rhodes, B.C.; Faigenbaum, A.D.; Myer, G.D. Relationships between functional movement screen scores, maturation and physical performance in young soccer players. J Sports Sci.2015, 33, 11–19.
Reviewer 2 Report
General comments
Congratulation for the statistical analyses with allometric modelling to identify optimal body size and shap characteristics associated with physical fitness tests in children and adolescents. Also, for the selection and explanation of the physical fitness tests.
Specific comments
Could you apport some information about the school/s (who many, main characteristics and type of the school/s, hours per week of physical education classes, socioeconomic status)? In terms of understanding better their characteristics.
Could you introduce some information about evaluators? Like always the same evaluators, experience evaluating this type of test. For example, in line 98-10: "anthropometric measurements were recorded in accordance with standardized procedures of the international society for the advancement of kinanthropometry (ISAK)". Have the evaluators any ISAK level?
As mentioned before, the present study aims to identify optimal body size and shape characteristics (using height, body mass and height-to-weight ratios) associated with nine physical fitness tests of Tunisian children and adolescents. In line75 or 79: " in a sample of non-specifically trained schoolchildren". How do you know they are untrained? Do you use any test, questionnaire?
In the line 89: The sample consisted of 597 Tunisian children (396 boys and 201 girls) aged 8-15 years. Can you explain what is the difference between gender?
Author Response
Dear reviewer,
We thank you for your valuable time and for the constructive and helpful comments. We carefully addressed all of your concerns and suggestions in the following point-by-point statement. Amendments to the manuscript were made whenever necessary.
Best Regards,
Dr. Helmi Chaabene (on behalf of all authors)
Reviewer 2 (changes were highlighted in GREEN)
General comments
Congratulation for the statistical analyses with allometric modelling to identify optimal body size and shape characteristics associated with physical fitness tests in children and adolescents. Also, for the selection and explanation of the physical fitness tests.
Authors’ response: Thank you for your positive feedback.
Comment 1
Could you apport some information about the school/s (who many, main characteristics and type of the school/s, hours per week of physical education classes, socioeconomic status)? In terms of understanding better their characteristics.
Authors’ response to comment 1: Thank you for your comment. In accordance with the reviewer’s suggestion, we provided more information on the respective schools, physical education classes, and study participants.
“Tests were always conducted during physical education classes (2 hours per week) by the same test instructor.”
“To be enrolled in this study, participants had to be apparently healthy (no diagnosed disease or injury) and not engaged in any structured training program. In this sense, children who were members of sport clubs were excluded from study participation. All schoolchildren were from families of low-to-medium socioeconomic status and enrolled in public elementary or high schools in an urban area of Tunisia.”
Comment 2
Could you introduce some information about evaluators? Like always the same evaluators, experience evaluating this type of test. For example, in line 98-10: "anthropometric measurements were recorded in accordance with standardized procedures of the international society for the advancement of kinanthropometry (ISAK)". Have the evaluators any ISAK level?
Authors’ response to comment 2: As requested by the reviewer, we have included more information on test instructors and instructions. None of the evaluators had an ISAK level. However, the inter-rater reliability as assessed by the intraclass correlation coefficient and standard error or measurement were > 0.90 and < 5%, respectively.
We have now added the following statement to the revised version of the manuscript:
“Tests were always conducted during physical education classes (2 hours per week) by the same test instructor.”
“All anthropometric measurements were recorded twice by two raters and the mean scores were retained for further statistical analysis. Inter-rater reliability was assessed using intraclass correlation coefficients (ICC) and error of measurement (SEM). For ICCs values were > 0.90 and for SEM < 5%.”
Comment 3:
As mentioned before, the present study aims to identify optimal body size and shape characteristics (using height, body mass and height-to-weight ratios) associated with nine physical fitness tests of Tunisian children and adolescents. In line75 or 79: " in a sample of non-specifically trained schoolchildren". How do you know they are untrained? Do you use any test, questionnaire?
Authors’ response to comment 3: Thank you for this comment. We did not assess physical activity directly or through questionnaire. However, we asked the children whether they were enrolled in any form of structured activities such as in a sport club. If this was the case, they were excluded from study participation.
“To be enrolled in this study, participants had to be apparently healthy (no diagnosed disease or injury) and not engaged in any structured training program. In this sense, children who were members of sport clubs were excluded from study participation.”
Comment 4:
In the line 89: The sample consisted of 597 Tunisian children (396 boys and 201 girls) aged 8-15 years. Can you explain what is the difference between gender?
Authors’ response to comment 4: As mentioned in your comment and the manuscript, 396 boys and 201 girls were enrolled in this study.
Reviewer 3 Report
The paper entitled Are early or late maturers likely to be fitter in the general population? aims to analyze to identify the optimal body-size/shape and maturity characteristics associated with superior fitness test performances having controlled for body-size, sex, and chronological-age differences.
The introduction of the paper was very descriptive, it did not situate the current study in literature or highlight what the gap in the literature is that this study is trying to address. In this sense, authors must provide better connections between variables under analysis. To my surprise I cannot understand, what is/are the theoretical model used in the present study, namely in terms of cognitive anxiety- In addition, the links between theoretical and empirical approaches are unclear, as well as the gaps of previous studies.
In relation to the contribution of the study to the literature, I did not get a sense from the article that the findings revealed anything other than what we already know. As I said before the introduction of the paper was very descriptive, it did not situate the current study in literature or highlight what the gap in the literature is that this study is trying to address.
Another concern is related to the literature gap. It is unclear what the gap that you intend to fill is?;
The hypothesis justification is very poor. Why do you believe in this hypothesis?
Overall, the introduction is too descriptive, lacks a clear argument for the study proposed, and does not situate the current study in the appropriate wider literature. Because of that, the authors must reorganize this section and clearly demonstrate strong arguments of the present study; situated the present study in literature, showed more previous studies about the relationships between variables under analysis.
The methods section lacks – General concerns
1- the recruitment date range (month and year);
2- a description of any inclusion/exclusion criteria that were applied to participant recruitment;
3- a table of relevant demographic details;
4- a statement as to whether your sample can be considered representative of a larger population,
5- a description of how participants were recruited, and descriptions of where participants were recruited and where the research took place.
6 – What is the power of the sample?
Statistical Analysis
In terms of statistical analysis, several informations all of the tests used must be clearly justified including all of the prerequisites.
Discussion
Overall, the discussion is very descriptive and any statements about the contribution and conclusions of the study are not new. What is the contribution to the literature, what is interesting about your results? The issues with the introduction about the lack of appropriate literature on the topic under analysis are replicated in the discussion.
How you can discuss theoretical implications if you do not approach the theoretical framework convincingly?
Lack of theoretical and empirical connection between the constructs analysed. The practical implications need to be further explored, as well as the limitations of the study.
Sometimes the discussion gives the impression that it is not very fluid and very descriptive (e.g., the results say this and what corroborates this and that), perhaps because there are few theoretical and empirical links about what is being analysed. In this sense, I suggest to the authors to make the discussion more fluid, organized into sub-topics and highlighting questions such as: why are your results important? What do you bring back to the literature?
Generally, the paper needs to be reviewed for spelling; grammar and punctuation could be improved in terms of the flow of the read. Finally, the manuscript was very difficult to read. Many of the sentences were extremely long and broken up by citations in multiple points. Simplifying the sentence structure would greatly improve the readability. Some aspects of the text were also quite repetitive.
Author Response
Dear reviewer,
We thank you for your valuable time and for the constructive and helpful comments. We carefully addressed all of your concerns and suggestions in the following point-by-point statement. Amendments to the manuscript were made whenever necessary.
Best Regards,
Dr. Helmi Chaabene (on behalf of all authors)
Reviewer 3 (changes were highlighted in GREY)
The paper entitled Are early or late maturers likely to be fitter in the general population? aims to analyze to identify the optimal body-size/shape and maturity characteristics associated with superior fitness test performances having controlled for body-size, sex, and chronological-age differences.
Comment 1: The introduction of the paper was very descriptive, it did not situate the current study in literature or highlight what the gap in the literature is that this study is trying to address. In this sense, authors must provide better connections between variables under analysis.
Authors’ response to comment 1: Thank you for your critical comment. We are sorry for not having been able to provide a convincing study rationale. Accordingly, the introduction was revised. We are confident that the gap in the literature is clearer now.
Comment 2:
To my surprise I cannot understand, what is/are the theoretical model used in the present study, namely in terms of cognitive anxiety- In addition, the links between theoretical and empirical approaches are unclear, as well as the gaps of previous studies.
Authors’ response to comment 2: Dear reviewer, we are currently unable to follow your argument in terms of ‘cognitive anxiety’. Please note that this was not the topic of our paper. We aimed to identify the optimal body-size/shape and maturity characteristics associated with superiour fitness test performances having controlled for body-size, sex, and chronological-age differences.
There is evidence in the literature (Lloyd et al, 2015; Lloyd et al, 2014) that growth, maturation, and sex have an impact on performance levels of different physical fitness qualities. With this study, we meant to examine the association between biological maturation and physical fitness as a continuous process in a sample of non-specifically trained schoolchildren.
Comment 3:
In relation to the contribution of the study to the literature, I did not get a sense from the article that the findings revealed anything other than what we already know. As I said before the introduction of the paper was very descriptive, it did not situate the current study in literature or highlight what the gap in the literature is that this study is trying to address.
Another concern is related to the literature gap. It is unclear what the gap that you intend to fill is?
Authors’ response to comment 3: Thank you for your critical but legitimate comment. Again, apologies for not having sufficiently described the gap in the literature. Most of the available studies on this topic focus on a cohort of trained children who were already enrolled in organized sports at the time of the study. Besides, age, maturation, and sex, training experience may also impact physical fitness performance levels. We purposefully focused on a sample of children that was not yet enrolled in organized sports to exclude the factor training experience from our analysis. The novelty of this study was that we focused on a sample of untrained youth. Moreover, we applied sophisticated statistical methods that allow an in-depth analysis of our data.
Comment 4:
The hypothesis justification is very poor. Why do you believe in this hypothesis?
Authors’ response to comment 4: In the introduction section, we mentioned that maturation is a non-linear (Towlson et al, 2018) process and there is ample evidence to support this hypothesis. The respective references have been cited in the introduction section. If the reviewer aims at a different issue we are currently not aware of, we would like to ask for more detailed information. Thank you.
Comment 5:
Overall, the introduction is too descriptive, lacks a clear argument for the study proposed, and does not situate the current study in the appropriate wider literature. Because of that, the authors must reorganize this section and clearly demonstrate strong arguments of the present study; situated the present study in literature, showed more previous studies about the relationships between variables under analysis.
Authors’ response to comment 5: See our responses to comments #1 and #2.
Comment 6:
The methods section lacks – General concerns
Authors’ response to comment 6: Dear reviewer, may we kindly ask you to provide further information on what exactly lacks in terms of the methods section? A prerequisite for an original research paper is that the study methods should be replicable. We provided detailed information for researchers and those interested in our work to replicate our study.
Comment 7:
The recruitment date range (month and year)
Authors’ response to comment 7: As requested by the reviewer, more information was provided in the revised version of the paper.
“This cross-sectional study was conducted between January to March 2018. It aimed at determining the influence of anthropometric characteristics, and biological maturation on PF variables in a sample of untrained schoolchildren”.
Comment 8
- a description of any inclusion/exclusion criteria that were applied to participant recruitment;
Authors’ response to comment 8: We have included additional information on inclusion/exclusion criteria.
“To be enrolled in this study, participants had to be apparently healthy (no diagnosed disease or injury) and not engaged in any structured training program. In this sense, children who were members of sport clubs were excluded from study participation. All schoolchildren were from families of low-to-medium socioeconomic status and enrolled in public elementary or high schools in an urban area of Tunisia.”
Comment 9
- a table of relevant demographic details;
Authors’ response to comment 9: Table 1 provides the information you asked for.
“Table 1. The number of children in the sample, plus the mean (±SD) body mass, height and maturation offset by sex and age group.”
Comment 10
4- a statement as to whether your sample can be considered representative of a larger population,
Authors’ response to comment 10: Our sample was a “purposive” sample, as defined below. The cohort is a fair representation of children from an urban area in Tunisia. A purposive sample is a non-probability sample that is selected based on characteristics of a population and the objective of the study. Purposive sampling is different from convenience sampling and is also known as judgmental, selective, or subjective sampling (Guarte et al., 2006).
Comment 11:
- a description of how participants were recruited, and descriptions of where participants were recruited and where the research took place.
Authors’ response to comment 11: Please refer to our response to comment 8.
Comment 12:
6 – What is the power of the sample?
Authors’ response to comment 12: Our sample was a “purposive” sample, as defined below. The cohort is a fair representation of children from an urban area in Tunisia. A purposive sample is a non-probability sample that is selected based on characteristics of a population and the objective of the study. Purposive sampling is different from convenience sampling and is also known as judgmental, selective, or subjective sampling (Guarte et al., 2006).
Comment 13:
In terms of statistical analysis, several informations all of the tests used must be clearly justified including all of the prerequisites.
Authors’ response to comment 13: Dear reviewer, the description of the statistics section contains the relevant information to replicate the statistical analyses and is in accordance with previous work from our research group (Nevill et al., 2009; Bustamante et al., (2015).
Comment 14
Overall, the discussion is very descriptive and any statements about the contribution and conclusions of the study are not new. What is the contribution to the literature, what is interesting about your results? The issues with the introduction about the lack of appropriate literature on the topic under analysis are replicated in the discussion.
How you can discuss theoretical implications if you do not approach the theoretical framework convincingly? Lack of theoretical and empirical connection between the constructs analysed.
Authors’ response to comment 14: We respectfully disagree with your comment. We discussed our study findings using the relevant literature and available theories in regards to factors that moderate physical fitness in youth. If the reviewer does not agree, we are happy to provide additional information during the next round of revisions. At the same time, we would like to ask the reviewer for more detailed information.
Comment 15:
The practical implications need to be further explored, as well as the limitations of the study.
Authors’ response to comment 15: Both sections have been revised. Thank you.
Comment 16
Sometimes the discussion gives the impression that it is not very fluid and very descriptive (e.g., the results say this and what corroborates this and that), perhaps because there are few theoretical and empirical links about what is being analysed. In this sense, I suggest to the authors to make the discussion more fluid, organized into sub-topics and highlighting questions such as: why are your results important? What do you bring back to the literature?
Authors’ response to comment 16: Dear reviewer, the discussion covers the most relevant literature (Iloyd et al, 2014; Iloyd et al; 2015) on the topic of physical fitness in youth. The relevance of our findings has been emphasized in the discussion. This study is novel in as much as we analyzed youth who were not organized in structured sports. Moreover, we applied sophisticated statistical methods that allow an in-depth analysis of our data.
Comment 17:
Generally, the paper needs to be reviewed for spelling; grammar and punctuation could be improved in terms of the flow of the read.
Authors’ response to comment 17: Three native speakers who are also coauthors of this paper proofread the article.
Comment 18
Finally, the manuscript was very difficult to read. Many of the sentences were extremely long and broken up by citations in multiple points. Simplifying the sentence structure would greatly improve the readability. Some aspects of the text were also quite repetitive.
Authors’ response to comment 18: We respectfully disagree with the reviewer. The paper was primarily written by native speakers. If the reviewer still disagrees, we kindly ask for details.
Round 2
Reviewer 1 Report
I would like to thank the authors for the substantial changes made throughout the manuscript. Most of my concerns were adequately answered in the new version.
Reviewer 3 Report
No more comments.